

# Exploring the use of a superconducting gravimeter to evaluate radar estimates of heavy rainfall

Laurent Delobbe[1], Arnaud Watlet[2], Svenja Wilfert[3], Michel Van Camp[2]

[1]Royal Meteorological Institute of Belgium, Brussels, B-1180, Belgium
[2]Royal Observatory of Belgium, Brussels, B-1180, Belgium
[3]Institute of Geography, University of Augsburg, Germany

*Correspondence to*: Laurent Delobbe (laurent.delobbe@meteo.be)

**Abstract.** The radar-based estimation of intense precipitation produced by convective storms is a challenging task and the verification through comparison with gauges is questionable due to the very high spatial variability of such type of
precipitation. In this study, we explore the potential benefit of using a superconducting gravimeter as a new source of in-situ observations for the evaluation of radar-based precipitation estimates. The superconducting gravimeter used in this study is installed in Membach (BE), 48 m underneath the surface, at 85 km distance from a C-band weather radar located in Wideumont (BE). The 15-year observation record 2003-2017 is available for both gravimeter and radar with 1-min and 5-min time steps, respectively. The gravimeter integrates soil water in a radius of about 400 m around the instrument. This allows capturing
rainfall at larger spatial scale than traditional rain gauges. The precision of the gravimeter is a few nm/s²; 1 nm/s² corresponding to 2.6 mm of water. The comparison of reflectivity and gravity time series for short duration intense rainfall events shows that reflectivity peaks larger than 40 dBZ are associated with a rapid decrease of the underground measured gravity. A remarkable correspondence between radar and gravimeter time series is found. The precipitation amounts derived from gravity measurements and from radar observations are further compared for 505 rainfall events. A correlation coefficient of 0.58, a
mean bias (radar/gravimeter) of 1.24 and a mean absolute difference (MAD) of 3.19 mm are obtained. A better agreement is reached when applying a hail correction by truncating reflectivity values to a given threshold. No bias, a correlation coefficient of 0.64 and a MAD of 2.3 mm are reached using a 48-dBZ threshold. The added value of underground gravity measurements as verification dataset is discussed. The two main benefits are the spatial scale at which precipitation is captured and the interesting property that gravity measurements are directly influenced by water mass at ground no matter the type of
precipitation: hail or rain.

## 1 Introduction

Weather radars are recognized as invaluable instruments for observing precipitation in the atmosphere. This is particularly true for precipitation fields exhibiting small scale patterns that cannot be easily captured by rain gauge networks. Due to their high



spatial and temporal resolutions, radar observations are crucial for the monitoring of such type of precipitation. However, radars only produce indirect rainfall estimates at ground derived from reflectivity measurements at several altitudes in the atmosphere. Rainfall intensities and amounts derived from radar observations are subject to numerous sources of uncertainties (e.g., Villarini and Krajewski, 2010; Berne and Krajewski, 2013). An evaluation of the quality of these estimations based on

verification datasets is therefore required. The verification of radar-based rainfall estimates is traditionally performed using rain gauge measurements. At a given location gauge measurements are relatively accurate but, unfortunately, not representative of a very large area due the high spatial variability of precipitation. Due to the limited network density, precipitation is only poorly monitored by rain gauges (Kidd et al., 2017). This poor representation is particularly marked for short-duration and local precipitation produced by convective storms (e.g. Schroeer et al., 2018). The lack of appropriate in-situ observations for

verification purpose does not only concern radar-derived precipitation but also satellite precipitation products. As mentioned in e.g. Puca et al. (2014), representativeness errors are introduced when comparing areal instantaneous data from satellites with punctual cumulated values from rain gauges. Similar challenges can be encountered for the validation of data from non-traditional sources like path-averaged precipitation from microwave links of cellular communication networks (Messer et al., 2006; Leijnse et al., 2007). A synthesis of future approaches for observing hydrological variables, including precipitation, is

presented in McCabe et al. (2017). The next decades will undoubtedly bring major advances in the observation of precipitation. However, the authors stress the importance of in-situ observations to support this progress by allowing the verification of rainfall inferred from new types of sensors and retrieval methods.

Some of the errors affecting radar precipitation estimates can be very large for heavy precipitation produced by convective

storms. For example, the conversion between radar reflectivity (Z) and rainfall intensity (R) is very uncertain in convective storms since the drop size distribution is extremely variable (e.g., Battan, 1973; Lee and Zawadzki, 2005). Besides, convective storms can produce precipitation in the form of hail, inducing a strong overestimation of radar-derived rainfall using commonly used Z-R relationships (Austin, 1987). Attenuation effects can also be particularly marked when intense rainfall is present between the radar and the location of interest (e.g., Delrieu et al., 2000). The temporal sampling, which is generally 5 minutes,

is also a limiting factor in the case of fast-moving small scale rainfall structures.

While uncertainties are large, the traditional approach for the validation of radar-derived rainfall based on comparison with gauges is particularly questionable in the case of convective precipitation. The difference of spatial representativeness between radar and gauge observations is indeed particularly problematic due to the large spatial variability of such type of precipitation.

A radar measurement is representative of a sample volume whose size increases with the distance to the radar. For a typical C-band weather radar with a 1 deg. resolution in azimuth and a range bin of 250m size, the projected area at ground lies between 0.04 km$^2$ at 10 km distance and 1 km$^2$ at 250 km. In contrast, a rain gauge collects precipitation over an interception area ranging from 100 to 500 cm$^2$ (Kidd et al., 2017).



As we have seen, the estimation and evaluation of extreme precipitation produced by convective storms is particularly challenging. In the context of a changing climate with an expected impact on the frequency of extreme rainfall (e.g. Ban et al., 2015; De Troch et al., 2013; Prein et al., 2016) , an accurate long-term monitoring of such type of precipitation is essential. Unfortunately, appropriate datasets are only poorly available for the verification of heavy convective rainfall. Given the

societal impact of heavy rainfall, it is necessary to explore alternative methods for evaluating radar-derived rainfall estimates based on new sources of independent rainfall observations.

In this study we explore the use of gravity measurements for this purpose. At the Earth surface, gravity ($g$) results from the attraction of the Earth, the Moon, the Sun and the centrifugal effects of the Earth rotation. When measuring the temporal

variations of the gravity field at a single location, three physical phenomena predominate in the signal: tides, atmospheric loading and polar motion of the Earth. Today, such tidal and polar motion effects can be easily removed from gravity measurements. State-of-the-art gravimeters are precise to better than the nm.s$^{-2}$ level ($10^{-10}$ $g$). At such a level, terrestrial gravimetric techniques allow monitoring local changes in the gravitational field associated with the variation of water masses present at the Earth surface. This results in the possibility to study local hydrological effects (Creutzfeldt et al., 2010a; Naujoks

et al., 2010), at a scale of up to 1 km², for signal ranging less than 1 nm/s² to a few thousands of nm/s² (Van Camp et al., 2017a). In the last two decades, gravity monitoring has been therefore increasingly used to study diverse kinds of hydrological processes such as soil moisture, rainfall, groundwater storage, hydrothermalism, or snow covering (Creutzfeldt et al., 2014; Hector et al., 2015; Hemmings et al., 2016; Imanishi et al., 2006; Jacob et al., 2008; Pool and Eychaner, 1995; Van Camp et al., 2006, 2016; Wilson et al., 2012). Surface instruments housed in buildings are often poorly sensitive to rainfall as an

umbrella effect limits the infiltration of rainwater into the ground in the area beneath the gravimeter (Creutzfeldt et al., 2010b; Deville et al., 2013). However, underground gravimeters are directly influenced by rainfall; in particular, intense rainfall events are clearly detected in gravity measurements (Meurers et al., 2007). For short-duration rainfall events, we can expect that hydro-meteorological processes like runoff, infiltration and evapotranspiration play a minor role and that gravity variations during the event are strongly related to rainfall amounts. A major advantage of underground gravity measurement is the spatial

scale which is much closer to the spatial scale of radar observations than gauges: 90% of the gravity signal caused by hydrological processes take place in a conic volume of radius r and height z, where r = 9.95 z (Singh, 1977). An underground superconducting gravimeter was installed in 1995 at Membach (Eastern Belgium). This gravimeter lies 48 m below the surface, which means that 90% of the gravity effect of rainfall is caused by water present within a radius of about 400 m around the gravimeter. This differs slightly from the r = 9.95 z relationship given the local topography (see supporting information in

Van Camp et al., 2016). Another advantage of gravimeter observation of precipitation is that it is not affected by the type of precipitation: rain, hail or snow. Gravity effects of precipitation are directly related to water mass.

Since 2002, concomitant time-series of superconducting gravimeter and weather radar measurements are available in Membach allowing us to explore the link between gravity and rainfall over 15 years. The goal of the present study is to use



these time-series (1) to identify and characterize the signature of intense rainfall events in gravity measurements and (2) to investigate the potential of gravimeters for evaluating the quality of radar-derived rainfall estimates. The radar and gravimeter data and the methods for deriving rainfall from these data are described in the next section. In section 3 we show that a rainfall signal is clearly visible in gravity time series and we compare radar-derived and gravity-derived rainfall amounts for a large

selection of short-duration intense rainfall events. Conclusions are presented in the last section.

## 2 Data and Methods

The data used in this study are produced by the Wideumont weather radar (49.9135°N, 5.5044°E) and the superconducting gravimeter (SG) GWR#C021 installed in 1995 at Membach (50.6085 N, 6.0095 E) (Van Camp et al., 2017b). The distance between Wideumont and Membach is 85.268 km (Fig. 1).

### 2.1 Gravimeter data

The fundamental component of a superconducting gravimeter, also called cryogenic gravimeter, consists in a hollow superconducting sphere that levitates in a persistent magnetic field generated by currents in a pair of superconducting coils (Goodkind, 1999; Hinderer et al., 2015). The superconducting property of zero resistance allows the currents that produce the magnetic field to flow forever without any resistive loss. Superconductivity is obtained by immersing the sensing unit in a

liquid helium bath at 4 K (269°C).

A change in gravity induces a vertical force on the sphere. As in modern spring gravimeters, the mass is kept at a constant position by injecting a current in an auxiliary feedback coil. Current SGs have a power spectral density noise level ranging typically 1–10 (nm/s²)²/Hz, which means that they are able to detect temporal gravity change ranging 0.1–0.3 nm/s2 (or 10–

30 nGal) within 1 min (Fores et al., 2017; Rosat and Hinderer, 2011; Van Camp et al., 2005). The weak instrumental drift of the SG, about 10 nm/s²/yr, is corrected using repeated absolute gravity measurements (Van Camp et al., 2017a). In this study, solid Earth and ocean loading effects were removed by computing tidal parameter sets using the ETERNA package (Wenzel, 1996) on the gravity time series extending from 1 June 2004 to 3 January 2015 (3825.75 record days). The tidal potential is the Hartmann-Wenzel (Hartmann and Wenzel, 1995) catalog with 7761 waves. The adjusted tidal parameters make it possible

to compute a tidal signal which includes both the solid Earth tide and ocean loading effects. The atmospheric loading effects were corrected by using a linear admittance factor also provided in the ETERNA package. It amounts to 3.3 nm/s²/hPa (Merriam, 1992), which means that a change of 1 hPa induces the same gravity change as produced by a 8.5 mm rainfall (see section 2.3). The local air pressure recording and a single admittance factor allow correcting for about 90% of the atmospheric effects (Boy, 2005; Boy et al., 2002, 2009; Hinderer et al., 2014; Klügel and Wziontek, 2009; Merriam, 1992). However, this

factor is frequency-dependent due to the spatial-temporal characteristics of pressure variations (Crossley et al., 2005; Wahr, 1985). Indeed, pressure fluctuations at short time scales are local and the impact on the gravity differs from the impact resulting





from slow pressure variations related to synoptic weather changes. At Membach for the period ranging 2005-2015, the coefficient decreases in average from -3.3 nm/s²/hPa at 1 cycle per day (cpd) to -3.8 nms⁻²/hPa at 10 cpd, then increases again up to -3.3 nm/s² at 36 cpd. Unfortunately, these values vary in time too, such that it is not possible to evaluate a frequency and time-dependent admittance. Hence we use in this study the admittance factor of -3.3 hPa/nm/s² classically used at the Membach

site. This implies that fluctuations in this factor are at the 15% level at worse.

The centrifugal effect associated with polar motion is also corrected (Wahr, 1985). The remaining gravity signal is usually called "residual". The residuals are corrected for undesirable element such as gap, steps, or spikes. These tares are essentially caused by maintenance and earthquakes (Hinderer et al., 2015). In the end of the processing chain, gravity residuals mainly

include the mixed effects of hydrological processes (both local and continental) and remaining tide and atmospheric pressure effects, which have not been perfectly corrected. Continental hydrological effects are at the seasonal scale and can be removed if needed using global hydrological models (e.g., Mikolaj et al., 2015). Conversely, local hydrological effects are at much higher frequency, up to the rain event scale (Meurers et al., 2007). The seasonal variations in the gravity signal are not of any concern when studying gravity variations at such a high frequency, which is the purpose of our study. The sampling rate is 60

15   s, after decimating and applying an anti-aliasing filter on the original 10 s-sampled data. In this study, precipitation amounts are derived from gravimeter data averaged over 5 minutes. At this period, the SG at the Membach station is able to monitor with a precision of 20 (nm/s²)²/Hz, corresponding to 0.2 nm/s² at a period of 300 s (Van Camp et al., 2005).

## 2.2 Radar data

The radar data used in this study have been produced by a C-band Doppler weather radar operated since 2002 by the Royal

Meteorological Institute of Belgium and located in Wideumont in the southeast of Belgium.  The radar is exploited for operational weather service but the observations have been also used in numerous research studies in meteorology, hydrology and ornithology (e.g., Goudenhoofdt et al., 2017; Dokter et al., 2011; Foresti et al., 2016).  Until 2015 the radar scanning strategy included a 5-elevation reflectivity scan every 5 min and a 10-elevation reflectivity scan every 15 minutes. Rainfall estimates were derived from the 5-min scan and hail detection was based on the 15-min scan. The scanning strategy changed

in December 2015. Ever since, rainfall and hail products have been inferred from a single full scan every 5 minutes. The scanning was originally performed bottom-up but it changed to top-down in 2015. It means that the lowest elevation rotation was first performed at the beginning of the 5-min cycle while it is now at the end of the 5-min cycle. The exact timestamp is used when comparing the 5-min or 15-min radar observations with the 1-min gravity measurements.

The radar beam width is 1 degree and the pulse length is 0.8 µs. The 5-min scan produces reflectivity data with a 1 degree resolution in azimuth and a 250 m resolution in range.  At 85 km distance, the main lobe is 1.48 km wide and the sample volume is 0.43 km³. The projected area at ground is 0.37 km² large, which is comparable to the 0.5 km² gravimeter sensitivity area. A Doppler filtering is applied to remove ground echoes. In this study the reflectivity data above Membach from the





lowest radar beam at 0.3 degree elevation are used. The height of these measurements is 1465 m a.s.l., which means 1171 m above ground level. It must be kept in mind that the radar measures instantaneous reflectivity at 5-minutes time intervals. Reflectivity (Z) data are converted into instantaneous rainrates (R) and rainfall amounts are further estimated through temporal integration. A hail detection method based on the Waldvogel method (Waldvogel et al., 1979; Delobbe and Holleman, 2006)

was used in this study to select severe convective events. The probability of hail is derived from the vertical profile of reflectivity and the freezing level.

### 2.3 Rainfall amounts from radar and gravimeter

Radar reflectivity values are converted into rain rates using the Marshall-Palmer (MP) relation, $Z=200 R^{1.6}$, which is the most commonly used Z-R relation (Marshall et al., 1955). The rainfall amount over 5 minutes (between -2.5 and +2.5 minutes) is

10 evaluated assuming that the rain rate is constant within that period of time. The cumulative rainfall is evaluating by summing the 5-min amounts. Cumulative rainfall is estimated from gravity measurements using the admittance factor of -0.39 nm s$^{-2}$/mm computed using a 1-m resolution digital elevation model (Van Camp et al., 2016), based on lidar data of the Public Service of Wallonia. It means that a gravity change of 1 nm s$^{-2}$ ($10^{-10}$ g) is produced by a 2.59 mm rainfall amount. Considering a precision of 0.2 nm s$^{-2}$, the lowest measurable rainfall amount is 0.5 mm. Assuming that gravity changes are

15 only due to precipitation the amount of precipitation can be evaluated by the gravity difference between two timestamps. Intense precipitation is expected to produce a gravity decrease. Gravity variations are also produced by other processes like infiltration and run off but we assume that, during the rainfall event, these effects are small with respect to the direct impact of rainfall at ground.

Small fluctuations at very short time scales (a few minutes) not related to precipitation are present in the gravity time series. Therefore, a 5-min temporal averaging of the gravity measurements has been applied for evaluating rainfall from gravity time series. The gravity change corresponding to a given radar reflectivity measurement at time t is taken as the difference between the mean gravity in the time intervals [t+2.5, t+7.5] and [t-7.5, t-2.5] expressed in minutes. For a full rainfall episode, which can last from a few minutes to a few hours, the associated gravity jump is calculated similarly based on the 5-min gravity

means before and after the episode.

The radar timestamp is not taken as the beginning or the end of the 5-min volume scan but as the time when the lowest radar sweep is located above the Membach station. The change in scanning strategy in December 2015, from bottom-up to top-down, is taken into account. A shift of the actual timestamp by 4 minutes is considered with respect to the nominal timestamp.

**2.4 Data selection and rainfall events**

The data selection is based on radar observations within the period 2003-2017. Less than 3 % of radar observations are missing within that period. A first explorative dataset has been produced by selecting days with severe convective precipitation. Severe




convective storms can produce hail and, therefore, the selection has been based on the radar-based hail detection. All days where the maximum probability of hail along the day exceeded 50 % at Membach station have been selected. This dataset includes 15 days for the whole time period. Such a small number of days is not surprising since, as shown in Lukach et al., (2017), the frequency of hail at a given location in Belgium is around 1 event per year. Among these 15 days, gravity data are

available for 14 days as a power outage made the SG data unavailable during the 2007-06-09 event. For these 14 days, the 5-min radar reflectivity time-series and the 1-min gravity measurements have been extracted and compared. In a second stage, a more extended radar dataset has been extracted based on radar reflectivity data only. All days where the maximum reflectivity along the day exceeds a given reflectivity threshold are extracted. For a reflectivity threshold of 40 dBZ (= 11 mm/h using MP), 408 days are extracted. Each day includes 288 data files, which represents more than 117 000 reflectivity measurements

above Membach station.

## 3 Results

The 14 convective days from the reduced dataset were used to get a first insight of the correspondence between gravity and reflectivity time series in case of very intense convective precipitation. The gravity and reflectivity time series for one of these

15    days (24 July 2017) are shown in Fig. 2. Several reflectivity peaks can be identified and the largest peaks are clearly associated with gravity changes. The two highest peaks are observed between 13:00 and 14:00 UTC and the corresponding 1 h gravity jump exceeds 4 nms$^{-2}$. The reflectivity values are further converted into rain rates and cumulative rainfall along the day. Cumulative rainfall is also estimated from gravity measurements using the admittance factor. The radar-derived rain rates and the radar- and gravimeter-derived cumulative rainfall are shown in Fig. 2 as well. A very good agreement is found between

the time series. Similar figures for all days are gathered in a supplement to this paper. Fig. 2 shows that the relation between radar reflectivity expressed in dBZ and rain rate is highly non-linear. Only very high reflectivity values correspond to heavy rainfall. A remarkable correspondence between the temporal evolution of radar and gravity measurements is generally found. The evolution of the atmospheric pressure at ground level along the day is shown in Fig. 3. The peaks in reflectivity and the corresponding gravity change between 13:00 and 14:00 UTC are associated with a 1 hPa pressure change. Considering an

error of 15% in the correction process, this means a maximum uncertainty of 0.5 nm/s², equivalent to 1.3 mm of water.

Figure 4 shows a scatter plot of the 5-min gravity change corresponding to the reflectivity data measured during the 14 selected days. 14x288 reflectivity measurements are included and for each measurement the gravity change is taken as the difference between the 5-min gravity mean before and after the measurement as described in previous section. Most of the observed

reflectivity values are less than 30 dBZ and do not show any signature in the gravity data. The 5-min variability of gravity in dry periods or in very light precipitation (less than 10 dBZ) is around 1 nm/s². Some signal is present for reflectivity larger than 30 dBZ and a clear effect of precipitation is observed when the reflectivity exceeds 40 dBZ, which corresponds to a rain rate of 11 mm/h, a 5-min rainfall amount of 0.9 mm using the MP relation, and a theoretical gravity change of 0.35 nm/s².





Even with very high reflectivity values, the 5-min rainfall amount remains relatively small. For example, a 55 dBZ value gives a 100 mm/h rain rate (using MP) and a resulting 5-min accumulation of 8.3 mm corresponding to a theoretical gravity change of 3.2 nm/s$^2$. In order to better evaluate gravity changes produced by large rainfall amounts it is interesting to analyse a large

number of rainfall events and to include events extending over several radar time steps. The extended dataset including 408 days with reflectivity larger than 40 dBZ above Membach is used for that purpose. Some of these days include more than one rainfall episode. In order to isolate intense rainfall events, consecutive measurements at least equal to 40 dBZ are grouped together to define one single rainfall event. When the time interval between successive events does not exceed 20 minutes, these events are regrouped as a same event. Using this procedure, we identify 563 intense rainfall events. Among these events

31 have been removed since the gravity data are affected by a power outage and 26 have been removed since the data are contaminated by an earthquake. An example of an earthquake can be seen on June 14 2006 around 5 UTC (Magnitude = 6.0, Aleutian islands, see Fig.8). In that particular case, the earthquake clearly occurs outside the rainy period and, therefore, the event is not eliminated.

Frequency distributions of event duration, pressure change and rainfall amounts characterizing the collection of 506 remaining events are shown in Fig. 5. For the rainfall amount, the frequency distribution is shown with two different frequency ranges. Almost all events have durations less than one hour and the radar-based rainfall amount is less than 10 mm in most cases. The atmospheric pressure change is determined following the same method as the gravity change. It is the difference between the 5-min mean pressure after and before the rainfall event. Even if rapid pressure changes can be observed within an intense

convective precipitation event, it appears that the atmospheric pressure before and after does not differ by more than 1 hPa in 95 % of the cases. The mean and standard deviation of the absolute pressure difference are 0.32 and 0.38 hPa, respectively. Considering an uncertainty of 15% on the admittance, this represents a maximal error of 0.5 nm/s² on the precipitation-induced gravity variation, which is equivalent to 1.3 mm of water.

A scatter plot of the gravity-based versus radar-based rainfall amounts based on the 506 events is shown in Fig. 6. The scatter plot shows a relatively good agreement between rainfall amounts. Table 1 gathers some statistics based on the 145 radar-gravimeter pairs with both values exceeding 2 mm. A Pearson's correlation coefficient of 0.58, a mean bias of 1.24 and a mean absolute difference of 3.2 mm are obtained. The mean bias is here defined as the ratio between the sum of all radar amounts and the sum of all gravimeter amounts. For large rainfall amounts the radar tends to overestimate with respect to the gravimeter.

Very high reflectivity values are generally observed during these events. These values might be produced by hail falls, which are known to produce substantially overestimated rainfall amounts when the classical MP Z-R conversion is used. The presence of hail stones in convective cells cause indeed a sharp increase in reflectivity with a relatively slight effect on the rainfall rate (Austin, 1987). In contrast, gravity measurements are not affected by the phase and the size of the hydrometeors. Only accumulated water mass determines the rainfall influence on the gravity.




A proper treatment of hail is recommended when producing quantitative precipitation estimates (QPE) from radar data. Conversion between reflectivity and equivalent rainrate in the case of hail or mixed rain-hail events is not straightforward and a simple correction is generally applied: all reflectivity values exceeding a given threshold are set to that threshold (e.g.

(Overeem et al., 2009). In the RMI QPE processing chain (Goudenhoofdt and Delobbe, 2016) , a reflectivity threshold of 55 dBZ is used and presented as a rather conservative value. The radar rainfall amounts have been recalculated using this truncation and a slightly better agreement between radar and gravimeter rainfall amounts is obtained with correlation coefficient and mean bias values of 0.60 and 1.20, respectively. Various thresholds values have been tested and it comes out that no bias is found between radar and gravimeter when a threshold of 48 dBZ is selected. The correlation coefficient reaches

then 0.64 and the mean absolute difference 2.33 mm. Figure 7 allows visualizing the effect of thresholding the reflectivity larger than 48 dBZ. The black points correspond to events where the maximum reflectivity does not exceed 48 dBZ and which are not affected by the hail correction. The radar-gravimeter pairs for the other events appear as red crosses and green squares, corresponding respectively to radar rainfall amounts without and with correction. The largest radar rainfall amount is obtained on June 14 2006 with 44 mm produced in a 40-min event. The amount obtained from gravity data is 9 mm. The hail detection

algorithm gives a probability of hail of 64 %. After correction the radar amount drops down to 22 mm. The temporal evolutions of gravity, reflectivity, rainfall rate and rainfall amount for this event are shown in Fig. 8. The results presented in Fig. 7 and Table 1 are consistent with the generally accepted view that the MP Z-R conversion tends to overestimate rainfall for very high reflectivity values and that some correction is required. The optimal 48-dBZ threshold found here might be influenced by other sources of uncertainties and should not be considered as a reference value that should be applied in any QPE processing.

However, our results suggest that a 55 dBZ thresholding applied before a MP Z-R conversion is insufficient to mitigate the radar rainfall overestimation associated with high reflectivity values generally produced by hail storms.

A large variety of Z-R conversion schemes have been proposed in the literature (e.g., Battan, 1973). In the RMI QPE processing scheme, the MP relation is used for reflectivity values below 44 dBZ while $Z=77 R^{1.9}$ is used for larger reflectivity values

following the DWD RADOLAN scheme (Wagner et al., 2012; Goudenhoofdt and Delobbe, 2016). Radar-based rainfall amounts have been evaluated using this Z-R conversion and the statistics characterizing the agreement between radar and gravimeter estimates are given in Table 1. The scores indicate that a correction for hail with a threshold close to 48 dBZ allows a better agreement between radar and gravimeter. The corresponding scatter plot is shown in Fig. 9. A bias very close to 1 is found between radar and gravimeter and, with respect to a pure MP conversion, the ZR conversion used in RMI QPE allows

a slight reduction of the MAD and the RMSE.





## 4. Conclusion

For the first time, observations from an underground superconductivity gravimeter and a C-band weather radar have been compared over 15 years for identifying and characterizing the signature of intense precipitation in gravity time series. Radar reflectivity data are converted into precipitation rates using the Marshall-Palmer relation and gravity data are converted using

an admittance factor of 0.39 nm s$^{-2}$/mm. A rainfall signal can be detected in gravity measurements once the radar reflectivity exceeds 40 dBZ. A surprisingly good correspondence is found between radar-derived and gravity-derived cumulative precipitation, especially as far as the temporal evolution of precipitation is concerned. Based on radar observations, 506 rainfall events with reflectivity exceeding 40 dBZ have been identified, among which 145 pairs where gravity- and radar- derived rainfall exceed 2 mm. Radar and gravimeter rainfall amounts have been compared and some statistics have been produced

based on these 145 radar-gravimeter pairs. A correlation coefficient of 0.58 and a mean bias (radar/gravimeter) of 1.24 are obtained. The precipitation overestimation of the radar with respect to the gravimeter is mainly due to very intense precipitation events characterized by very high radar reflectivity values. Hail is often produced by such storms and our results show that applying a hail correction by truncating reflectivity to a given threshold allows a substantial improvement of the agreement between radar and gravity precipitation amounts. Best results are obtained with a 48-dBZ threshold, which is lower than the

55-dBZ commonly used threshold. The atmospheric pressure near the gravimeter is measured in order to correct for the atmospheric effects on the gravity. It appears that for 95 % of the precipitation events, the pressure difference before and after the event does not exceed 1 hPa. This result is important since it means that errors in gravity-derived rainfall amounts caused by inadequate correction of pressure effects hardly exceed 1 mm.

In the present study, we have shown the benefit of using gravimeter observations for the verification of radar-derived precipitation amounts. The essential added value of precipitation estimates derived from underground gravimeters with respect to traditional rain gauges is the spatial scale at which precipitation is captured. The gravimeter at Membach is sensitive to precipitation falling within a radius of 400 m around the station. A single gravimeter captures actually more precipitation than the 400 stations of the rain gauge networks in Belgium. The spatial representativeness is of course totally different. The

gravimeter can be seen as a spatial integrator of precipitation producing observations at ground which much better match weather radar observations than rain gauges. The temporal sampling of precipitation by gravimeter (less than 1 minute) is also fully appropriate for hydro-meteorological applications. Another advantage of gravimeter-derived precipitation observations stems from the measurement principle directly based on the mass of precipitation at ground. For a given mass per square meter, liquid water, snow or hail have the same influence on gravity. In contrast, weather radar observations are strongly affected by

the microphysical properties of precipitation, in particular the phase and the size distribution of hydrometeors. Rain gauge measurements are also affected by various errors which depend on the type of precipitation (rain, hail or snow). In case of very short and intense hail falls we can expect the best correspondence between gravity changes and precipitation amounts. Indeed, run off and infiltration processes are generally slower, which limits their impact during such events. An accurate evaluation of

precipitation amounts in case of extreme precipitation, possibly with hail, is essential since radar observations are more and more used to derive extreme rainfall statistics. Gravimeter observations allow to point out and, up to a certain extent, to quantify the overestimation of rainfall extremes by the radar due to hail. We conclude that gravimeter can help improving rainfall estimates in case of hail and very intense rain. On the other hand, as a complement to rain gauges, radar provides valuable

information for routine detection of sudden changes in gravity time series. This is important for the analysis of geodynamical signals such as tides, Earth's free oscillations or slow tectonic deformations.

In the present study we focused on rain and hail fall events producing large precipitation amounts over short durations. For longer events with moderate precipitation, evapotranspiration, run off and infiltration are expected to produce a larger effect

on gravity changes. The joint analysis or radar and gravity time series in such rainfall events can also bring valuable information for further studies in hydrology and hydrogeology.

## Acknowledgements

We thank the technical teams at RMI and ROB for the careful maintenance of the Membach gravimeter station and the

Wideumont weather radar. Comments and suggestions provided by Maarten Reyniers (RMI) were greatly appreciated.

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





| Z-R relation | Hail correction | Pearson's corr. coef. | Pearson's P_value | Mean BIAS | MAD (mm) | RMSE (mm) |
|---|---|---|---|---|---|---|
| Marshall-Palmer | No correction | 0.58 | 2.2E-14 | 1.24 | 3.19 | 5.44 |
| | Z threshold = 55 dBZ | 0.60 | 1.7E-15 | 1.20 | 2.94 | 4.81 |
| | Z threshold = 48 dBZ | 0.64 | 2.4E-18 | 1.00 | 2.33 | 3.29 |
| RMI QPE | No correction | 0.61 | 6.5E-16 | 1.15 | 2.78 | 4.45 |
| | Z threshold = 55 dBZ | 0.62 | 1.6E-16 | 1.13 | 2.64 | 4.13 |
| | Z threshold = 48 dBZ | 0.65 | 3.9E-18 | 0.97 | 2.32 | 3.23 |

5 **Table 1: statistics based on 145 valid pairs with radar and gravimeter rainfall amounts both exceeding 2 mm. Mean bias = sum(Ri)/sum(Gi), where Ri and Gi are the radar and gravimeter rainfall amounts. MAD is the mean absolute difference and RMSE the root mean square error.**





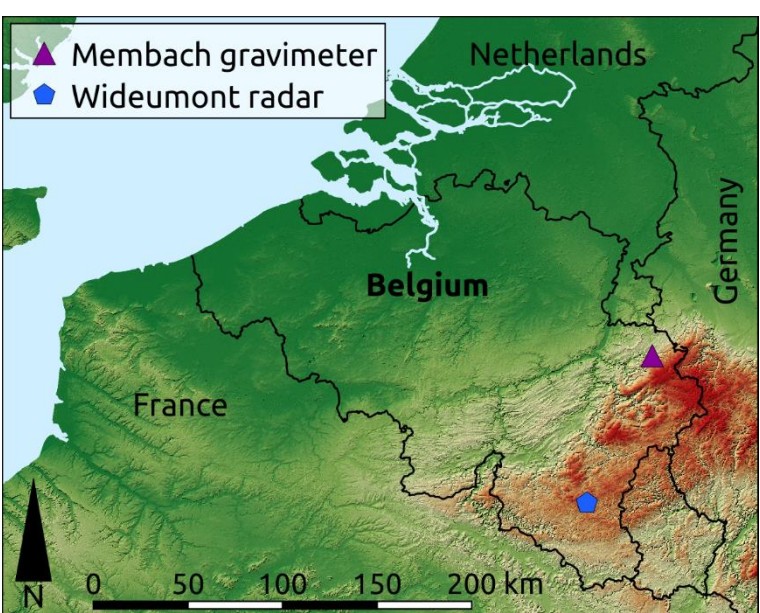

**Figure 1: Locations of gravimeter and weather radar.**





**Figure 2: Time series for 2017/07/24 0-24 UTC: residual gravity (nm s⁻²), radar reflectivity (dBZ), radar-derived rainfall rate (mm/h) and cumulative rainfall (mm) derived from gravity and radar.**





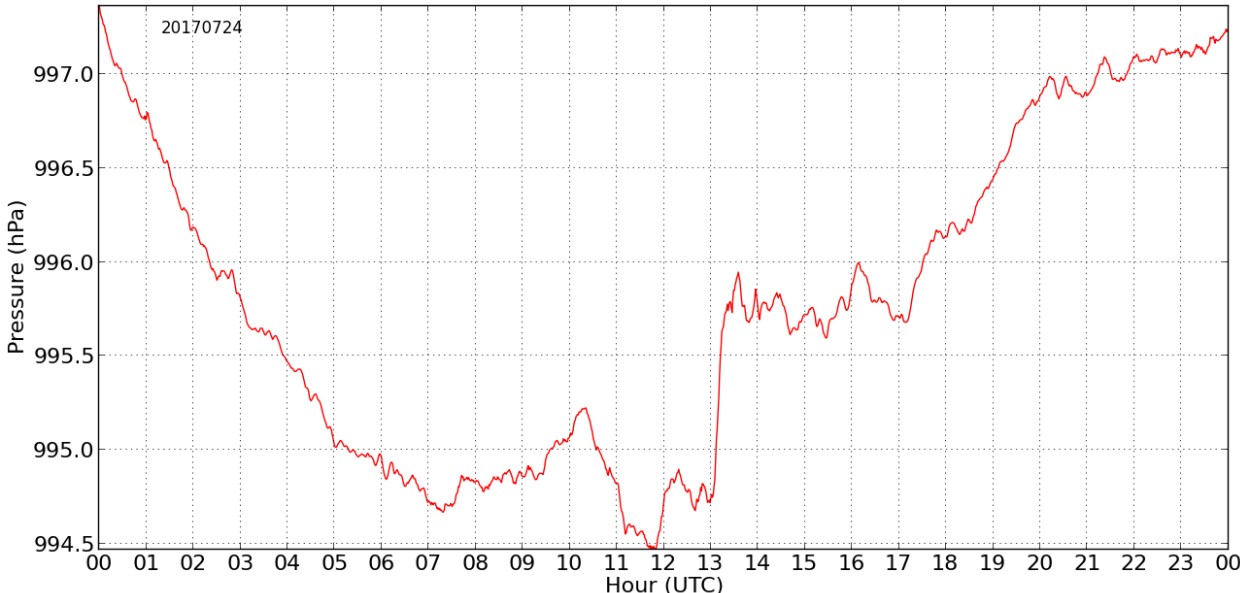

**Figure 3: Time series for 2017/07/24 0-24 UTC : ground atmospheric pressure (hPa).**





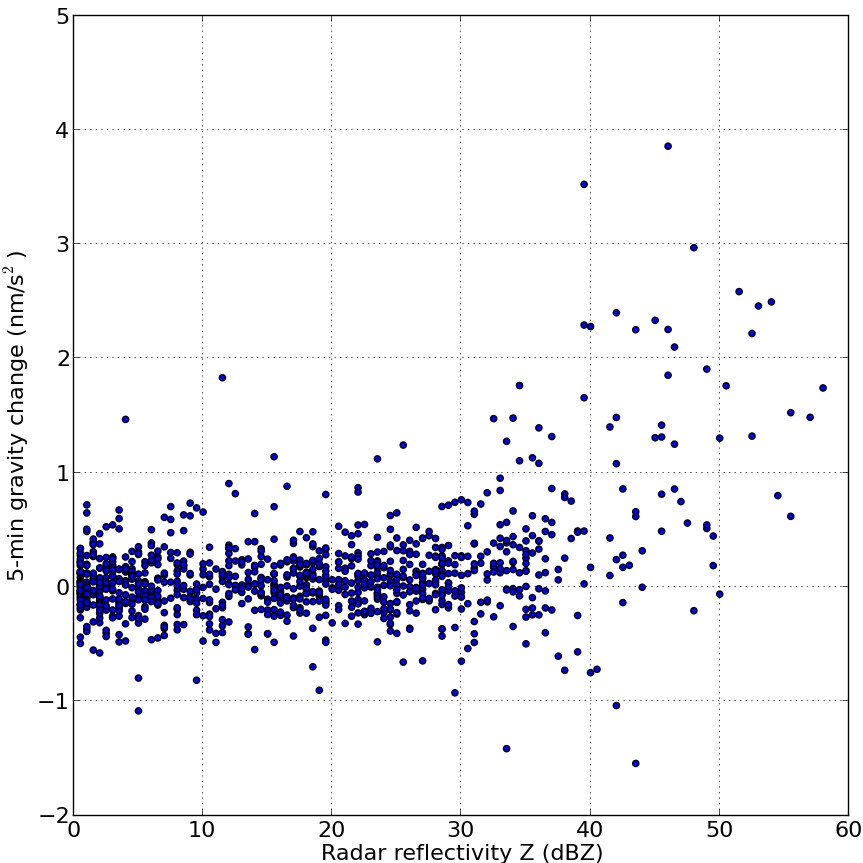

Figure 4: 5-min gravity change as function of radar reflectivity based on 14 days with heavy rainfall.







**Figure 5: Frequency distribution of duration, pressure change and radar rainfall amounts (with 2 different frequency maximum ranges) based on 506 intense rainfall events.**





**Figure 6: Rainfall amounts derived from radar and gravimeter measurements for 506 precipitation events with max reflectivity exceeding 40 dBZ.**



**Figure 7: Rainfall amounts derived from radar and gravimeter measurements for 506 precipitation events with max reflectivity exceeding 40 dBZ (black points). The MP relation is used and a hail correction using a 48-dBZ threshold is applied. The red crosses and the green squares correspond to radar rainfall amounts without and with hail correction, respectively. Black points correspond to the pairs which are not affected by hail correction.**







**Fig. 8: Residual gravity, radar reflectivity, radar rainfall rate and cumulative rainfall time series for 2006/06/14, not corrected for hail (red) and corrected with a hail threshold of 48 dBZ (green).**







**Figure 9: Rainfall amounts derived from radar and gravimeter measurements for 506 precipitation events with max reflectivity exceeding 40 dBZ (black points). The ZR relation from RMI QPE is used and a hail correction using a 48-dBZ threshold is applied.**

