# Peer review of "Exploring the use of underground gravity monitoring to evaluate radar estimates of heavy rainfall"

_Hydrology and Earth System Sciences, 2018_

## Referee Comment (RC1) · H. Leijnse (Referee) · 25 Sep 2018

In this paper rainfall estimates from a weather radar and an underground gravimeter are compared. A straightforward method of estimating rainfall from the gravimeter is presented. Gravity measurements are first corrected for tidal effects and atmospheric loading, and are then converted to rain accumulations by applying a moving average, and a linear relation between gravity and accumulated water. Results that are presented show that the gravimeter indeed has a strong precipitation signal. It is also shown that the gravimeter data can help in the case of hail, where radars typically over-estimate precipitation rates. The paper is well-written and very interesting for readers

of HESS. As far as I know, this is the first time that rainfall estimates from an underground gravimeter are reported, and hence the paper is very novel. The paper could benefit from adding known information about uncertainties in gravimeter data in order to facilitate comparison of the two. I have a few further suggestions for minor changes to the paper, after which I think the paper is ready for publication. Specific remarks are given below.

**Specific remarks**

1. In Figs 2, 4, and 6-9, please add error margins to the gravimeter data so that it is immediately apparent what the expected uncertainty of these measurements are.

2. On p.6, line 23, it is stated that the 5-minute gravity change is averaged over 5-minute intervals in order to remove high-frequency gravity fluctuations due to other sources. It would be very interesting to know how this 5-minute time scale is related to the time scales of hydrological processes that would affect the gravity measurements through the redistribution of water. Please add a statement on the typical timescales of these processes. This can then be related to the 5-minute scale of averaging, but also to the typical time scales of individual rainfall events.

3. In Fig.2, it is clear that there are high-frequency fluctuations present in the gravimeter data. I would recommend discussing these fluctuations, and potential ways to remove them. For example, would it be possible to average gravity differences over longer time intervals (say, 15 or 30 minutes) to remove most of these fluctuations? It would be interesting to see the effect of different averaging time scales on this apparent noise. Please consider adding a sensitivity analysis to the scale of averaging.

4. On p.10, lines 5-6, it is concluded that a rainfall signal can be detected when radar reflectivity exceeds 40 dBZ. However, this conclusion is based on comparison of the 5-minute signals. I think that it is very well possible to detect rainfall signals even if reflectivity values are lower than this if the rainfall is averaged over sufficiently long time periods. This is for example demonstrated in Fig.2, between 1:00 and 6:00 UTC, where the radar only exceeds 20 dBZ most of the time, and 30 dBZ on just a few occasions. Yet the total accumulation by the gravimeter nicely follows that of the radar. So I think that this conclusion is too hard on the method that is presented. Please rephrase this conclusion to reflect this.

5. On p.10, lines 14-15, it is concluded that a 48-dBZ hail cap works better than a 55-dBZ hail cap. This 48-dBZ hail cap corresponds to 36 mm h$^{-1}$ (for M-P) or 34 mm h$^{-1}$ (for RADOLAN). These thresholds seem rather low, and could result in missing significant rainfall. My guess is that this optimal threshold is partially a result of compensation for error sources other than hail. Please comment about this in the paper.

6. In order to make the paper more concise, I suggest removing Figs 6 and 9. The points in Fig.6 are already given in Fig.7, and there is only a minor difference between Figs 7 and 9. Furthermore, results from Fig.9 are also summarized in Table 1.

**Minor remarks**

1. On p.1, the title does not include the fact that the gravimeter is underground, but this is an essential element of the paper (it probably would no work so well if the gravimeter was at the surface). Please modify the title to reflect this. Suggestion: replace "superconducting" by "underground"

2. On p.2, line 12, consider replacing "puntual" by "point-scale".

3. On p.2, line 23, consider mentioning that this is mostly the case for C- and X-band radars (not so much for S-band radars).

4. On p.2, line 25, consider referring to Fabry et al. (1994; https://doi.org/10.1016/0022-1694(94)90138-4)

5. On p.2, line 33, consider adding a remark that the radar sampling area is at least 1 million times as large as a gauge sampling area.

6. On p.5, lines 3 and 4, there's a typo in the units (should be "nm/s$^2$/hPa"; this occurs twice: on lines 3 and 4). Consider using using notations for units such as "nm s$^{-2}$ hPa$^{-1}$" instead of using the "/" character throughout the paper to increase readability.

7. On p.5, lines 3-5, it is mentioned here that the values of the coefficients vary with with time as well as the frequency of pressure fluctuations. In the conclusion that is drawn that there is a 15% uncertainty in the gravimeter data is based on the differences between the minimum and maximum values of these coefficients. It is hence implicitly assumed that the time variations of these values is much less than the variation with frequency (or cpd). Is this correct? If so, please add a statement that the time variation is much smaller than the variation with cpd.

8. On p.5, line 8, what are "tares"? I assume this refers to the "gap, steps, or spikes" from the previous sentence. Consider removing this word such that it reads "These are essentially...".

9. On p.5, line 17, it is unclear to me what "precision" means in this context. Is this the noise expressed in the power spectral density of the gravity signal? Or is it something else (such as mentioned in the abstract on p.1, line 15). Please briefly mention in the paper what is meant by the precision here.

10. On p.5, line 32, consider removing the word "large".

11. On p.8, line 23, the mean bias is defined as the ratio of the radar sum and the gravimeter sum. I generally interpret "bias" to mean the systematic error, that becomes negative in case of underestimation (and I think more readers would, too). I therefore recommend expressing the bias as the sum of differences divided by the sum of the reference (i.e., the radar). In practice, this means subtracting 1 from the original numbers. But, in my view, it does give more clarity.

12. On p.8, line 30, consider removing the word "falls".

13. On p.10, line 32, consider removing the word "falls".

14. On p.11, line 8, consider removing the word "fall".

---

## Referee Comment (RC2) · Anonymous Referee #2 · 18 Oct 2018

General Comments

This paper presents a study comparing precipitation estimates derived from weather radar observations and gravimeter data. This is an original contribution to the field dealing with a topic of scientific significance, the estimation of the precipitation, from a non-traditional approach. The comparison methods are relatively standard and the results and conclusions well justified. A few clarifications and suggestions that may enhance the quality of the manuscript are listed below, along with some minor formal corrections.

Specific Comments

[Figure]

1. Page 2, line 10. Considering TRMM or GPM spaceborne weather radars I think this should be modified: radar-derived -> ground-based radar derived

2. Pages 5 (line 30) - Page 6 (line 1). "The radar beam width is 1 degree ... the lowest radar beam at 0.3 degree elevation is used". If the radar half beam-width is 1 degree, then using a 0.3 degree antenna elevation does not imply substantial beam blockage? Unless the radar antenna is higher than the surrounding terrain.. I think this should be briefly explained in the text.

3. Page 6, line 16. "Intense precipitation is expected to produce a gravity decrease". This is a crucial point of the paper and, perhaps because it is very obvious for the authors, it is only mentioned very briefly. In my opinion this sentence deserves a longer explanation, perhaps one or two additional sentences.

4. Page 6, line 29 (last sentence of section 2.3). Why a 4 minute shift in the timestamp is considered? Please explain briefly (or perhaps simply connect with the previous sentences).

5. Pages 6-7, section Data Selection. The weather radar used operates at C-band so attenuation with heavy precipitation and/or hail is a potential problem. When selecting the events, did authors consider identifying and discarding attenuation cases by checking the radar sector (or specific radials) which extends from the radar site to the gravimeter site? I think this should be commented.

6. Page 9. I found interesting the analysis described where different radar reflectivity thresholds are applied for QPE conversion. The values reported are consistent with those used for QPE estimates in the US National Mosaic and Multi-Sensor QPE (NMQ) system - see Zhang et al (2011), p. 1329 - where different capping dBZ values are considered for pixels classified as convective, warm-rain and hail: 55, 50 and 49 dBZ respectively. I think this could be further commented.

Technical Comments

7. Page 1, line 17 (and elsewhere where amounts are considered). Suggest: larger than -> greater than

8. Page 4, line 19. Please check meaning: change -> changes?

9. Page 4, line 19. Typo: check superindex in nm/s2 -> nm/s[super_index]2 OR nm s[super_index]-2

10. Page 6, line 10. Please check meaning: is evaluating -> is evaluated

11. Page 8, line 12. Figure 8 is cited after Figure 4 and before Figure 5. Please consider reordering/renumbering the figures to cite them in order.

REFERENCE

Zhang, J., et al. (2011). National Mosaic and Multi-Sensor QPE (NMQ) system: Description, results, and future plans. Bulletin of the American Meteorological Society, 92(10), 1321-1338.

---

## Author Comment (AC1) · 25 Oct 2018

**Answers to review by Hidde Leijnse (reviewer's comments in black, answers in blue)**

In this paper rainfall estimates from a weather radar and an underground gravimeter are compared. A straightforward method of estimating rainfall from the gravimeter is presented. Gravity measurements are first corrected for tidal effects and atmospheric loading, and are then converted to rain accumulations by applying a moving average, and a linear relation between gravity and accumulated water. Results that are presented show that the gravimeter indeed has a strong precipitation signal. It is also shown that the gravimeter data can help in the case of hail, where radars typically overestimate precipitation rates. The paper is well-written and very interesting for readers of HESS. As far as I know, this is the first time that rainfall estimates from an underground gravimeter are reported, and hence the paper is very novel. The paper could benefit from adding known information about uncertainties in gravimeter data in order to facilitate comparison of the two. I have a few further suggestions for minor changes to the paper, after which I think the paper is ready for publication. Specific remarks are given below.

The authors would like to thank Hidde Leijnse for the very positive comments and the very relevant remarks and suggestions.

Uncertainties in gravimeter data are already described in the paper but we will add more explicit information on these uncertainties in a revised version of our paper.

Specific remarks

1. In Figs 2, 4, and 6-9, please add error margins to the gravimeter data so that it is immediately apparent what the expected uncertainty of these measurements are

For the time series (Figs 2 and 8) we propose to use a rolling standard deviation to plot a 2-sigma margin on the data. For the 5-min gravity changes in Fig. 4 and in event gravity changes in fig 6-7-9 (scatter plots) we propose to take the average standard deviation before and after the event. This requires some re-processing. We need also to check whether adding uncertainty information in the scatter plots will not reduce the readability of the figures.

2. On p.6, line 23, it is stated that the 5-minute gravity change is averaged over 5-minute intervals in order to remove high-frequency gravity fluctuations due to other sources. It would be very interesting to know how this 5-minute time scale is related to the time scales of hydrological processes that would affect the gravity measurements through the redistribution of water. Please add a statement on the typical timescales of these processes. This can then be related to the 5-minute scale of averaging, but also to the typical time scales of individual rainfall events.

We chose to average gravity data at 5-min scale in order to match radar data. As shown in Fig. 5, most rainfall events are very short (less than 15 minutes). Besides, when evaluating the change in gravity produced by such events, it is reasonable to average the gravity over short time intervals as well. Averaging the gravity, for example, over one hour and evaluating the gravity change by taking the 1h-hour average before the event and after the event would incorporate in the gravity changes various effects which are not directly related to precipitation, e.g. pressure changes, ground-water processes, …

Concerning the timescales, we have seen that the intense rainfall events analyzed here occur over timescales of minutes. As recommended by the reviewer, this should be compared with timescales of other processes affecting the redistribution of water. Evapotranspiration occurs typically at diurnal timescales (2-3 mm/day during sunny summer days, Van Camp et al., GRL 2016) and infiltration at timescales of hours. Characteristic timescales of runoff are more difficult to quantify and depend on the status of the soil saturation.  We think that runoff is the predominant process affecting gravity at timescales close to those of precipitation. However, we find in most cases that gravity does not substantially and rapidly increase after the rainfall events, which suggests that runoff is slower than precipitation process.

3. In Fig.2, it is clear that there are high-frequency fluctuations present in the gravimeter data. I would recommend discussing these fluctuations, and potential ways to remove them. For example, would it be possible to average gravity differences over longer time intervals (say, 15 or 30 minutes) to remove most of these fluctuations? It would be interesting to see the effect of different averaging time scales on this apparent noise. Please consider adding a sensitivity analysis to the scale of averaging.

We use data with a one-minute sampling rate : it means that periods longer than 120 s were filtered out. Then, looking at the power spectral densities of SGs, the noise reaches a flat, lower level at periods ranging 50-500 s. Hence, averaging on 300 s (or 5 minutes) allows benefiting from the period at which the SG presents the best performances.

At longer periods, the noise becomes red (increasing power with increasing period). Hence, for rainfalls of 30-60 minutes, concurrently to the rain-induced drop in gravity, there is another signal induced by the red noise. However, this red noise, at those periods, contains essentially the hydrological (e.g., Van Camp et al., JGR 2010) and atmospheric effects. The hydrological signal is investigated here, while we assess the effect of the atmospheric effects by discussing the pressure signal before and after the event. Therefore, we think that presenting a sensitivity study to the scale of averaging is beyond the scope of the present study.

4. On p.10, lines 5-6, it is concluded that a rainfall signal can be detected when radar reflectivity exceeds 40 dBZ. However, this conclusion is based on comparison of the 5-minute signals. I think that it is very well possible to detect rainfall signals even if reflectivity values are lower than this if the rainfall is averaged over sufficiently long time periods. This is for example demonstrated in Fig.2, between 1:00 and 6:00 UTC, where the radar only exceeds 20 dBZ most of the time, and 30 dBZ on just a few occasions. Yet the total accumulation by the gravimeter nicely follows that of the radar. So I think that this conclusion is too hard on the method that is presented. Please rephrase this conclusion to reflect this.

This is a very relevant remark. We found indeed several moderate rainfall episodes where the radar reflectivity does not exceed 40 dBZ but where a very good correspondence is found between the cumulated rainfall derived from the radar and from the gravimeter along the day. However, there are also many cases where we don't find any correspondence when only moderate rainfall is observed. In the latter cases, it seems that hydrogeological processes dominate over direct precipitation effect. It is worth mentioning that gravity fluctuations due to hydrogeological processes are likely to be dependent on the seasons and status of the soil moisture saturation (Van Camp et al., JGR 2006), which has to be done. This a very interesting research topic which requires additional research. It is already mentioned in the last paragraph of the conclusion but we will elaborate a bit more on this in a revised version.

Coming back to the reviewer's comment, we agree that it is not justified to conclude that no gravity signal is observed when the radar reflectivity is lower than 40 dBZ. This will be adapted in the revised paper.

5. On p.10, lines 14-15, it is concluded that a 48-dBZ hail cap works better than a 55-dBZ hail cap. This 48-dBZ hail cap corresponds to 36 mm h−1 (for M-P) or 34 mm h−1 (for RADOLAN). These thresholds seem rather low, and could result in missing significant rainfall. My guess is that this optimal threshold is partially a result of compensation for error sources other than hail. Please comment about this in the paper.

It is well known that hail produces rainfall overestimation when the M-P relation is used but, even for rain, overestimation can occur. This is due, for example, to radar electronic miscalibration, ZR relation that is not adapted to real drop size distribution, or partial evaporation of rainfall below the height of the measurement (VPR effect). In page 9 line 18-19 we mention that the 48-bBZ can be influenced by other sources uncertainties. As underlined by the reviewer, hail cannot be blamed as the only responsible for radar overestimations. We will elaborate more on this in the revised version, taking also into account the comment given by the second reviewer.

6. In order to make the paper more concise, I suggest removing Figs 6 and 9. The points in Fig.6 are already given in Fig.7, and there is only a minor difference between Figs 7 and 9. Furthermore, results from Fig.9 are also summarized in Table 1.

There is indeed some redundant information in figures 6 and 7. Concerning Fig. 9 the aim is to show the best agreement that we get between radar- and gravimeter-derived rainfall amounts. It is a simple figure that can be easily extracted from the paper and used by the scientific community to summarize in one slide the use of gravity measurements to derive rainfall amounts. Fig. 6 can be used for that purpose as well. Therefore, we would propose to remove Fig. 6 or Fig. 9 but not both.

Minor remarks

1. On p.1, the title does not include the fact that the gravimeter is underground, but this is an essential element of the paper (it probably would no work so well if the gravimeter was at the surface). Please modify the title to reflect this. Suggestion:

replace "superconducting" by "underground"

We propose to use "Exploring the use of underground gravity monitoring to evaluate radar estimates of heavy rainfall" as a new title.

Gravimeters installed underground are usually sensitive to a large area at the surface, which improves their rainfall detection. Conversely, gravimeters installed at ground level generally sample a much smaller portion of the top layer, but this varies from site to site. On the one hand, the buildings hosting gravimeters installed at ground level act as an umbrella, which prevents direct infiltration of rainwater below the gravimeter. It has been proved that such a building mask reduces significantly the signal recorded after rainfall events (Creutzfedldt et al., 2008). Reducing the size of the building, which hosts a gravimeter, as well as raising the gravimeter on a pillar tends to mitigate such building mask effects, as shown by Güntner et al. (2017). On the other hand, specific topography settings can also favor rainfall to be detected by surface gravimeters. This is for example the case of gravimeters installed at the top of a hill. However, underground gravimeters are in any case much more suitable to study rainfall processes because they are not concerned by building mask effects and much less affected by topography effects, while sampling larger areas of the surface layer.

References:

Creutzfeldt et al., 2008 https://library.seg.org/doi/10.1190/1.2992508

Guntner et al., 2017 https://www.hydrol-earth-syst-sci.net/21/3167/2017/

2. On p.2, line 12, consider replacing "puntual" by "point-scale".

Will be done.

3. On p.2, line 23, consider mentioning that this is mostly the case for C- and X-band radars (not so much for S-band radars).

Will be done.

4. On p.2, line 25, consider referring to Fabry et al. (1994; https://doi.org/10.1016/0022-1694(94)90138-4)

Will be done.

5. On p.2, line 33, consider adding a remark that the radar sampling area is at least 1 million times as large as a gauge sampling area.

Very good suggestion. The comparison between sampling areas of the radar, the gravimeter, and a rain gauge is very instructive. Will be done.

6. On p.5, lines 3 and 4, there's a typo in the units (should be "nm/s2/hPa"; this occurs twice: on lines 3 and 4). Consider using using notations for units such as "nm s−2 hPa−1" instead of using the "/" character throughout the paper to increase readability.

Will be done.

7. On p.5, lines 3-5, it is mentioned here that the values of the coefficients vary with time as well as the frequency of pressure fluctuations. In the conclusion that is drawn that there is a 15% uncertainty in the gravimeter data is based on the differences between the minimum and maximum values of these coefficients. It is hence implicitly assumed that the time variations of these values is much less than the variation with frequency (or cpd). Is this correct? If so, please add a statement that the time variation is much smaller than the variation with cpd.

We will add this statement.

8. On p.5, line 8, what are "tares"? I assume this refers to the "gap, steps, or spikes" from the previous sentence. Consider removing this word such that it reads "These are essentially...".

Tares refers indeed to all kinds of anomalies. It is frequently used in gravimetric jargon. We will remove it from this paper, clearly mentioning gaps, steps and spikes.

9. On p.5, line 17, it is unclear to me what "precision" means in this context. Is this the noise expressed in the power spectral density of the gravity signal? Or is it something else (such as mentioned in the abstract on p.1, line 15). Please briefly mention in the paper what is meant by the precision here.

We propose to replace :

"At this period, the SG at the Membach station is able to monitor with a precision of 20 (nm/s²)²/Hz, corresponding to 0.2 nm/s² at a period of 300 s (Van Camp et al., 2005)."

by

"The power spectral density of the SG at the Membach station is at the level of 20 (nm/s²)²/Hz, which corresponds to a precision of 0.2 nm/s² at a period of 300 s (Van Camp et al., 2005)."

and, in the abstract, to replace

"The precision of the gravimeter is a few nm/s²;"

by

"The precision of the gravimeter is a few tenths of nm/s²;"

10. On p.5, line 32, consider removing the word "large".

Will be done.

11. On p.8, line 23, the mean bias is defined as the ratio of the radar sum and the gravimeter sum. I generally interpret "bias" to mean the systematic error, that becomes negative in case of underestimation (and I think more readers would, too). I therefore recommend expressing the bias as the sum of differences divided by the sum of the reference (i.e., the radar). In practice, this means subtracting 1 from the original numbers. But, in my view, it does give more clarity.

Good suggestion.

12. On p.8, line 30, consider removing the word "falls".

Will be done.

13. On p.10, line 32, consider removing the word "falls".

Will be done.

14. On p.11, line 8, consider removing the word "fall".

Will be done.

---

## Author Comment (AC2) · 25 Oct 2018

**Answers to anonymous referee #2 (referee's comments in black, answers in blue)**

General Comments

This paper presents a study comparing precipitation estimates derived from weather radar observations and gravimeter data. This is an original contribution to the field dealing with a topic of scientific significance, the estimation of the precipitation, from a non-traditional approach. The comparison methods are relatively standard and the results and conclusions well justified. A few clarifications and suggestions that may enhance the quality of the manuscript are listed below, along with some minor formal corrections.

The authors thank the reviewer for the very positive feedback and the interesting suggestions.

Specific Comments

1. Page 2, line 10. Considering TRMM or GPM spaceborne weather radars I think this should be modified: radar-derived -> ground-based radar derived

Good remark. Will be adapted.

2. Pages 5 (line 30) - Page 6 (line 1). "The radar beam width is 1 degree ... the lowest radar beam at 0.3 degree elevation is used". If the radar half beam-width is 1 degree, then using a 0.3 degree antenna elevation does not imply substantial beam blockage? Unless the radar antenna is higher than the surrounding terrain. I think this should be briefly explained in the text.

The radar antenna is installed on top of a 50-m tower. The surroundings are lower than the antenna and the beam blockage is very limited. This especially true in the direction of Membach, the location of interest where the gravimeter is installed. This will be briefly explained in the text.

3. Page 6, line 16. "Intense precipitation is expected to produce a gravity decrease". This is a crucial point of the paper and, perhaps because it is very obvious for the authors, it is only mentioned very briefly. In my opinion this sentence deserves a longer explanation, perhaps one or two additional sentences.

Since the gravimeter is underground, the increase of water mass at ground level due to precipitation results in a decrease of the measured gravity. The fact that the gravimeter is underground is an essential characteristic, because (1) the sensitivity radius reaches a few hundreds of meters and (2) there is no building preventing rainwater from being measured ("umbrella effect"). However, in some cases, gravimeters installed at the surface have already been used to study soil moisture processes, as shown by Guntner et al 2017 in which their superconducting gravimeter is installed in a small field enclosure.

As recommended by reviewer 1, we will reformulate the manuscript and the title to make it more explicit since the very beginning. As a new title, we propose: "*Exploring the use of underground gravity monitoring to evaluate radar estimates of heavy rainfall*"

4. Page 6, line 29 (last sentence of section 2.3). Why a 4 minute shift in the timestamp is considered? Please explain briefly (or perhaps simply connect with the previous sentences).

When the 3D scanning of the atmosphere is performed starting from the highest elevation angle, 4 minutes are approximately necessary to reach the lowest elevation angle after 14 antenna rotations. This will be explained in the revised manuscript.

5. Pages 6-7, section Data Selection. The weather radar used operates at C-band so attenuation with heavy precipitation and/or hail is a potential problem. When selecting the events, did authors consider identifying and discarding attenuation cases by checking the radar sector (or specific radials) which extends from the radar site to the gravimeter site? I think this should be commented.

No selection has been performed based on attenuation effects between the radar and the location of interest. It means that rainfall underestimations are possible when heavy rain or hail is present in the corresponding radial. This will be commented in the revised version.

6. Page 9. I found interesting the analysis described where different radar reflectivity thresholds are applied for QPE conversion. The values reported are consistent with those used for QPE estimates in the US National Mosaic and Multi-Sensor QPE (NMQ) system - see Zhang et al (2011), p. 1329 - where different capping dBZ values are for pixels classified as convective, warm-rain and hail: 55, 50 and 49 dBZ respectively. I think this could be further commented.

This is a very interesting comment. We were not aware that the optimal threshold found in our study was consistent with the 49-dBZ threshold used for capping dBZ values in hail as described in Zhang et al. (2011). It is interesting to note that the capping value used for rain (55 dBZ) is substantially larger than the one used for hail. When capping all reflectivity values to 48 dBZ as we do, we certainly underestimate some very intense precipitation in the form of rain. This issue was raised by reviewer 1. We will extend the discussion of our results in the revised version.

Technical Comments

7. Page 1, line 17 (and elsewhere where amounts are considered). Suggest: larger than -> greater than.

Will be done.

8. Page 4, line 19. Please check meaning: change -> changes?

Will be done.

9. Page 4, line 19. Typo: check superindex in nm/s2 -> nm/s[super_index]2 OR nm s[super_index]-2

Will be checked and adapted throughout the text. This was also mentioned by reviewer 1.

10. Page 6, line 10. Please check meaning: is evaluating -> is evaluated

Will be adapted.

11. Page 8, line 12. Figure 8 is cited after Figure 4 and before Figure 5. Please consider reordering/renumbering the figures to cite them in order.

Will be adapted.

**Reference**

Guntner et al 2017 https://www.hydrol-earth-syst-sci.net/21/3167/2017/

---

## Author Response (AR1)

**Answers to review by Hidde Leijnse (reviewer's comments in black, answers in red)**

In this paper rainfall estimates from a weather radar and an underground gravimeter are compared. A straightforward method of estimating rainfall from the gravimeter is presented. Gravity measurements are first corrected for tidal effects and atmospheric loading, and are then converted to rain accumulations by applying a moving average, and a linear relation between gravity and accumulated water. Results that are presented show that the gravimeter indeed has a strong precipitation signal. It is also shown that the gravimeter data can help in the case of hail, where radars typically overestimate precipitation rates. The paper is well-written and very interesting for readers of HESS. As far as I know, this is the first time that rainfall estimates from an underground gravimeter are reported, and hence the paper is very novel. The paper could benefit from adding known information about uncertainties in gravimeter data in order to facilitate comparison of the two. I have a few further suggestions for minor changes to the paper, after which I think the paper is ready for publication. Specific remarks are given below.

The authors would like to thank Hidde Leijnse for the very positive comments and the very relevant remarks and suggestions.

Uncertainties in gravimeter data are better described in the revised paper and error margins have been added on figures 2, 4, 6 and 8.  Additional information and explanations on the uncertainties can be found in the revised version as described below.

As recommended by Hidde Leijnse, the tithe has been changed. The new title is : "Exploring the use of underground gravity monitoring to evaluate radar estimates of heavy rainfall".

All modifications brought to meet the comments provided by Hidde Leijnse are visible in red in the revised version included at the end of the present document. The page and line numbers mentioned in our responses refer to the version with changes highlighted.

Specific remarks

1. In Figs 2, 4, and 6-9, please add error margins to the gravimeter data so that it is immediately apparent what the expected uncertainty of these measurements are

Additional information and explanations on the uncertainties can be found at several places throughout the text. (p4 L29,  p5 L18,  p5 L33,  p8 L22 L31 , p9 L29).

Error margins have been added taking into account the uncertainty of the gravity measurements at 5-min time scale and the uncertainty of the pressure correction. This pressure correction aims at correcting the impact of the atmospheric mass.

For the temporal evolution (Fig. 2 and 8), we consider a characteristic pressure variation of 1 hPa. For Fig. 4 and 6, the real pressure variation is used for each rainfall event. The error bars are not included in Fig. 7 since it makes the figure too heavy while the uncertainty information is already shown in Fig. 6.

Our original plan, as explained during the interactive discussion, was to plot a rolling standard deviation but it produces a very small error margin which does not include the impact of the pressure correction.

2. On p.6, line 23, it is stated that the 5-minute gravity change is averaged over 5-minute intervals in order to remove high-frequency gravity fluctuations due to other sources. It would be very interesting to know how this 5-minute time scale is related to the time scales of hydrological processes that would affect the gravity measurements through the redistribution of water. Please add a statement on the typical timescales of these processes. This can then be related to the 5-minute scale of averaging, but also to the typical time scales of individual rainfall events.

We chose to average gravity data at 5-min scale in order to match radar data. As shown in Fig. 5, most rainfall events are very short (less than 15 minutes). Besides, when evaluating the change in gravity produced by such events, it is reasonable to average the gravity over short time intervals as well. Averaging the gravity, for example, over one hour and evaluating the gravity change by taking the 1h-hour average before the event and after the event would incorporate in the gravity changes various effects which are not directly related to precipitation, e.g. pressure changes, ground-water processes, …

Concerning the timescales, we have seen that the intense rainfall events analyzed here occur over timescales of minutes. As recommended by the reviewer, this should be compared with timescales of other processes affecting the redistribution of water. Evapotranspiration occurs typically at diurnal timescales (2-3 mm/day during sunny summer days, Van Camp et al., GRL 2016) and infiltration at timescales of hours. Characteristic timescales of runoff are more difficult to quantify and depend on the status of the soil saturation. We think that runoff is the predominant process affecting gravity at timescales close to those of precipitation. However, we find in most cases that gravity does not substantially and rapidly increase after the rainfall events, which suggests that runoff is slower than precipitation process.

The text includes now this new information/discussion. See p7 L4 to L20.

3. In Fig.2, it is clear that there are high-frequency fluctuations present in the gravimeter data. I would recommend discussing these fluctuations, and potential ways to remove them. For example, would it be possible to average gravity differences over longer time intervals (say, 15 or 30 minutes) to remove most of these fluctuations? It would be interesting to see the effect of different averaging time scales on this apparent noise. Please consider adding a sensitivity analysis to the scale of averaging.

We use data with a one-minute sampling rate : it means that periods longer than 120 s were filtered out. Then, looking at the power spectral densities of SGs, the noise reaches a flat, lower level at periods ranging 50-500 s. Hence, averaging on 300 s (or 5 minutes) allows benefiting from the period at which the SG presents the best performances.

At longer periods, the noise becomes red (increasing power with increasing period). Hence, for rainfalls of 30-60 minutes, concurrently to the rain-induced drop in gravity, there is another signal induced by the red noise. However, this red noise, at those periods, contains essentially the hydrological (e.g., Van Camp et al., JGR 2010) and atmospheric effects. The hydrological signal is investigated here, while we assess the effect of the atmospheric effects by discussing the pressure signal before and after the event. Therefore, we think that presenting a sensitivity study to the scale of averaging is beyond the scope of the present study.

4. On p.10, lines 5-6, it is concluded that a rainfall signal can be detected when radar reflectivity exceeds 40 dBZ. However, this conclusion is based on comparison of the 5-minute signals. I think that it is very well possible to detect rainfall signals even if reflectivity values are lower than this if the rainfall is averaged over sufficiently long time periods. This is for example demonstrated in Fig.2, between 1:00 and 6:00 UTC, where the radar only exceeds 20 dBZ most of the time, and 30 dBZ on just a few occasions. Yet the total accumulation by the gravimeter nicely follows that of the radar. So I think that this conclusion is too hard on the method that is presented. Please rephrase this conclusion to reflect this.

This is a very relevant remark. We found indeed several moderate rainfall episodes where the radar reflectivity does not exceed 40 dBZ but where a very good correspondence is found between the cumulated rainfall derived from the radar and from the gravimeter along the day. However, there are also many cases where we don't find any correspondence when only moderate rainfall is observed. In the latter cases, it seems that hydrogeological processes dominate over direct precipitation effect. It is worth mentioning that gravity fluctuations due to hydrogeological processes are likely to be dependent on the seasons and status of the soil moisture saturation (Van Camp et al., JGR 2006), which has to be done. This a very interesting research topic which requires additional research, as mentioned in the last paragraph of the conclusion.

Coming back to the reviewer's comment, we agree that it is not justified to conclude that no gravity signal is observed when the radar reflectivity is lower than 40 dBZ. This has been adapted in the revised paper. See abstract L17 and conclusion L17.

5. On p.10, lines 14-15, it is concluded that a 48-dBZ hail cap works better than a 55-dBZ hail cap. This 48-dBZ hail cap corresponds to 36 mm h−1 (for M-P) or 34 mm h−1 (for RADOLAN). These thresholds seem rather low, and could result in missing significant rainfall. My guess is that this optimal threshold is partially a result of compensation for error sources other than hail. Please comment about this in the paper.

It is well known that hail produces rainfall overestimation when the M-P relation is used but, even for rain, overestimation can occur. This is due, for example, to radar electronic miscalibration, ZR relation that is not adapted to real drop size distribution, or partial evaporation of rainfall below the height of the measurement (VPR effect). In page 9 line 18-19 we mention that the 48-bBZ can be influenced by other sources uncertainties. As underlined by the reviewer, hail cannot be blamed as the only responsible for radar overestimations.

A new paragraph has been added in the revised version at the end of section 3 taking also into account the comment given by the second reviewer (p11 L1 to L11)

6. In order to make the paper more concise, I suggest removing Figs 6 and 9. The points in Fig.6 are already given in Fig.7, and there is only a minor difference between Figs 7 and 9. Furthermore, results from Fig.9 are also summarized in Table 1.

There was indeed some redundant information in figures 6 and 7. In the revised manuscript, error margins have been added to figure 6 but not in figure 7 to keep it readable. Fig. 7 aims at showing the impact of the hail correction. The two figures are now complementary. Figure 9 has been moved to the supplement.

Minor remarks

1. On p.1, the title does not include the fact that the gravimeter is underground, but this is an essential element of the paper (it probably would no work so well if the gravimeter was at the surface). Please modify the title to reflect this. Suggestion:

replace "superconducting" by "underground"

The title has been changed. We use "Exploring the use of underground gravity monitoring to evaluate radar estimates of heavy rainfall" as a new title.

Gravimeters installed underground are usually sensitive to a large area at the surface, which improves their rainfall detection. Conversely, gravimeters installed at ground level generally sample a much smaller portion of the top layer, but this varies from site to site. On the one hand, the buildings hosting gravimeters installed at ground level act as an umbrella, which prevents direct infiltration of rainwater below the gravimeter. It has been proved that such a building mask reduces significantly the signal recorded after rainfall events (Creutzfedldt et al., 2008). Reducing the size of the building, which hosts a gravimeter, as well as raising the gravimeter on a pillar tends to mitigate such building mask effects, as shown by Güntner et al. (2017). On the other hand, specific topography settings can also favor rainfall to be detected by surface gravimeters. This is for example the case of gravimeters installed at the top of a hill. However, underground gravimeters are in any case much more suitable to study rainfall processes because they are not concerned by building mask effects and much less affected by topography effects, while sampling larger areas of the surface layer.

Has be done.

**Answers to anonymous referee #2 (referee's comments in black, answers in blue)**

General Comments

This paper presents a study comparing precipitation estimates derived from weather radar observations and gravimeter data. This is an original contribution to the field dealing with a topic of scientific significance, the estimation of the precipitation, from a non-traditional approach. The comparison methods are relatively standard and the results and conclusions well justified. A few clarifications and suggestions that may enhance the quality of the manuscript are listed below, along with some minor formal corrections.

The authors thank the reviewer for the very positive feedback and the interesting suggestions. All comments have been taken into account to prepare the revised version.

All modifications brought to meet the comments provided by the second reviewer are visible in blue in the revised version with changes highlighted at the end of the present document.

The page and line numbers mentioned in our responses refer to the version with changes highlighted.

Specific Comments

1. Page 2, line 10. Considering TRMM or GPM spaceborne weather radars I think this should be modified: radar-derived -> ground-based radar derived

Good remark. Has been adapted (p2 L13).

2. Pages 5 (line 30) - Page 6 (line 1). "The radar beam width is 1 degree ... the lowest radar beam at 0.3 degree elevation is used". If the radar half beam-width is 1 degree, then using a 0.3 degree antenna elevation does not imply substantial beam blockage? Unless the radar antenna is higher than the surrounding terrain. I think this should be briefly explained in the text.

The radar antenna is installed on top of a 50-m tower. The surroundings are lower than the antenna and the beam blockage is very limited. This especially true in the direction of Membach, the location of interest where the gravimeter is installed. This is now briefly explained in the text (p6 L6)

3. Page 6, line 16. "Intense precipitation is expected to produce a gravity decrease". This is a crucial point of the paper and, perhaps because it is very obvious for the authors, it is only mentioned very briefly. In my opinion this sentence deserves a longer explanation, perhaps one or two additional sentences.

Since the gravimeter is underground, the increase of water mass at ground level due to precipitation results in a decrease of the measured gravity. As long as rain or hail is in the atmosphere, its effect on gravity is corrected based on local air pressure measurement. In contrast, water mass on ground has a direct impact on the measured gravity.  This is now better explained in the text (see abstract L14, p3 L28, p5 L6)

The fact that the gravimeter is underground is an essential characteristic, because (1) the sensitivity radius reaches a few hundreds of meters and (2) there is no building preventing rainwater from being measured ("umbrella effect"). However, in some cases, gravimeters installed at the surface have already been used to study soil moisture processes, as shown by Guntner et al 2017 in which their superconducting gravimeter is installed in a small field enclosure.

As recommended by reviewer 1, we will reformulate the manuscript and the title to make more explicit since the very beginning that the gravimeter is underground. As a new title, we propose: "*Exploring the use of underground gravity monitoring to evaluate radar estimates of heavy rainfall*"

4. Page 6, line 29 (last sentence of section 2.3). Why a 4 minute shift in the timestamp is considered? Please explain briefly (or perhaps simply connect with the previous sentences).

When the 3D scanning of the atmosphere is performed starting from the highest elevation angle, 4 minutes are approximately necessary to reach the lowest elevation angle after 14 antenna rotations. This is explained in the revised manuscript (p7 L25).

5. Pages 6-7, section Data Selection. The weather radar used operates at C-band so attenuation with heavy precipitation and/or hail is a potential problem. When selecting the events, did authors consider identifying and discarding attenuation cases by checking the radar sector (or specific radials) which extends from the radar site to the gravimeter site? I think this should be commented.

No selection has been performed based on attenuation effects between the radar and the location of interest. It means that rainfall underestimations are possible when heavy rain or hail is present in the corresponding radial. This is mentioned in the revised version (p8 L6).

6. Page 9. I found interesting the analysis described where different radar reflectivity thresholds are applied for QPE conversion. The values reported are consistent with those used for QPE estimates in the US National Mosaic and Multi-Sensor QPE (NMQ) system - see Zhang et al (2011), p. 1329 - where different capping dBZ values are for pixels classified as convective, warm-rain and hail: 55, 50 and 49 dBZ respectively. I think this could be further commented.

This is a very interesting comment. We were not aware that the optimal threshold found in our study was consistent with the 49-dBZ threshold used for capping dBZ values in hail as described in Zhang et al. (2011). It is interesting to note that the capping value used for rain (55 dBZ) is substantially larger than the one used for hail. When capping all reflectivity values to 48 dBZ as we do, we certainly underestimate some very intense precipitation in the form of rain. This issue was raised by reviewer 1. The discussion of our results has been extended in the revised version. See last paragraph of section 3, p11)

Technical Comments

7. Page 1, line 17 (and elsewhere where amounts are considered). Suggest: larger than -> greater than.

The whole sentence has been revised following recommendation by reviewer 1.

8. Page 4, line 19. Please check meaning: change -> changes?

Has been done.

9. Page 4, line 19. Typo: check superindex in nm/s2 -> nm/s[super_index]2 OR nm s[super_index]-2

Ha been adapted throughout the text. This was also mentioned by reviewer 1.

10. Page 6, line 10. Please check meaning: is evaluating -> is evaluated

Has been adapted.

11. Page 8, line 12. Figure 8 is cited after Figure 4 and before Figure 5. Please consider reordering/renumbering the figures to cite them in order.

Reference to the figure has been modified.

**Reference**

Guntner et al 2017 https://www.hydrol-earth-syst-sci.net/21/3167/2017/

[revised manuscript text omitted]

FIG 9 MOVED TO SUPPLEMENT